# Time-Dependent Cytokine-Release of Platelet-Rich Plasma in 3-Chamber Co-Culture Device and Conventional Culture Well

**Chih-Hao Chiu** [1,*] **, Poyu Chen** [1,2] **, Alvin Chao-Yu Chen** [3] **, Yi-Sheng Chan** [3] **, Kuo-Yao Hsu** [3] **and Kin Fong Lei** [4,*]

[1] Department of Orthopedic Surgery, Taoyuan Chang Gung Memorial Hospital, Taoyuan 333, Taiwan; poyuchen@mail.cgu.edu.tw

[2] Department of Occupational Therapy and Graduate Institute of Behavioral Sciences, College of Medicine, Chang Gung University, Taoyuan 333, Taiwan

[3] Department of Orthopedic Surgery, Linkou Chang Gung Memorial Hospital, Taoyuan 333, Taiwan; alvin_taiwan@outlook.com (A.C.-Y.C.); yschan512@gmail.com (Y.-S.C.); emsequoia@cgmh.org.tw (K.-Y.H.)

[4] Graduate Institute of Biomedical Engineering, Chang Gung University, Taoyuan 333, Taiwan

\* Correspondence: joechiu0115@gmail.com (C.-H.C.); kflei@mail.cgu.edu.tw (K.F.L.)

**Abstract:** Platelet-rich plasma (PRP) contains bioactive cytokines to enhance tissue healing. The best PRP preparation protocol and timing of the treatment have not been determined yet. To screen the best-fit PRP, a 3-chamber co-culture device was developed. We hypothesized the concentrations of the cytokines from different PRPs in the co-culture plates had a high correlation with those in conventional 24-well culture plates at different time points. The concentrations of the cytokine from PRPs would be correlated with platelet concentrations. The correlation of transforming growth factor beta-1 (TGF-β1) and platelet-derived growth factor AB (PDGF-AB) in both devices were compared at 0, 24, 48, 72, and 96 h from two PRPs as well as that of platelet and cytokines concentrations. The results revealed that there was a moderate to high correlation in TGF-β1 concentrations between the 3-chamber co-culture and conventional culture device until 96 h. The correlation of PDGF-AB concentrations in both devices had moderate to high correlation in the first 24 h, and then it became modestly correlated from 48 to 96 h. A high correlation was found between platelet and TGF-β1 concentration at 96 h. However, they were modestly correlated in other time points. A negative or modest correlation was found between platelet and PDGF-AB concentration in all time points. In conclusion, TGF-β1 and PDGF-AB revealed a time-dependent manner of release at five time points. There is a moderate to high correlation of the TGF-β1 and PDGF-AB concentration in both devices at different time points. However, TGF-β1 and PDGF-AB concentrations are not always proportional to the platelet concentration of the PRPs.

**Keywords:** platelet-rich plasma; co-culture; TGF-β1; PDGF-AB; time-dependent



## 1. Introduction

Platelet-rich plasma (PRP) is an autologous blood-derived product that is applied exogenously to various tissues. After platelet activation, high concentrations of platelet-derived cytokines such as transforming growth factor beta-1 (TGF-β1) and platelet-derived growth factor (PDGF) that enhance tissue healing are released [1–3]. TGF-β1 is believed to be a key growth factor to stimulate collagen synthesis, cell proliferation, recruitment, and migration [4]. PDGF was shown to upregulate proteoglycan synthesis and acts as a chemotactic factor for cells of a mesenchymal origin [5].

PRP is used to promote wound healing, although its use still raises various controversies related to the effectiveness of platelet concentrates in tissue regeneration [6]. Moreover, there were mixed results in PRP's clinical efficacy [7,8]. Such inconsistencies were attributed to differences in established preparation protocols [9] so as to the timing and dosing frequency of PRP to treat different kinds or stages of diseases [10]. More than 40 commercial systems claim to concentrate whole blood into PRP, but a standardized preparation protocol

has yet to be implemented [9]. It is also unclear why PRP preparation is considered the optimal treatment for various cell types and that a "more is better" theory for the use of higher platelet concentrations could not be supported. This is because the stimulating effects of PRP may turn into the opposite beyond an ideal dosage [11,12]. Furthermore, the platelet count is not always corresponding to the cytokine content for some individual preparations, such as different isolation methods, centrifuge speed, and activation methods [13]. Many studies using different PRP preparations and application protocols have produced various results because of the dose-dependent and time-dependent relationship between the platelets/cytokines delivered to the injury site [14]. It has yet to be established at what time point these cytokines should be addressed clinically to provide optimal effects on tissue healing.

To screen the best-fit PRP preparation protocol to a specific tissue, a specially-designed platform of a 3-chamber co-culture device was developed [15]. The main idea of this system is to avoid the gelling effect of PRPs [16,17] and to decrease the cell amount needed to find out its relationship with different PRPs because the diseased tissue harvested during surgery was sometimes very limited. Freshly cultured cells were not manifold available in a sufficient amount [18]. Tenocytes, for example, lose partial phenotype as early as the third passages [19]. Only cells within the first three passages should be used for the study [20]. With this device, less amount and earlier passages of cells could be co-cultured with different PRPs to find out their best interaction with each other when compared with using conventional culture wells with the larger culture diameter.

Therefore, we hypothesized that: (1) The concentration of cytokines from different PRPs in the 3-chamber co-culture device had a high correlation with those in the conventional culture wells at different time points; (2) The cytokine concentrations in different PRPs would be correlated with platelet concentrations. Such information could help clinicians to determine the in vitro best PRP preparation protocol by using the co-culture device without taking too much specimen during surgery. Second, the different concentrations of cytokines from PRPs at different time points could provide help to determine the optimal timing of PRPs applications in clinical practice.

## 2. Results

### 2.1. Subjects

Blood samples were obtained from 19 patients (10 women, 9 men; mean age $\pm$ SD, 58.4 $\pm$ 16.3 years; range, 26–65 years). The study protocol was approved by the institutional review board at our institution. Subjects had to be healthy, between the ages of 20 and 65, and without any known blood dyscrasia. The exclusion criteria comprised any form of anticoagulant, antibacterial, or immunosuppressive therapy within the last 6 months, any form of systemic illness, and current or recent history of cancer [21].

### 2.1.1. Mean Blood Characteristics from 19 Patients

The mean platelet counts of the whole blood samples from 19 patients was 205.3 $\pm$ 50.7 $\times$ 10$^3$ cells/$\mu$L. The platelet counts after the second spin were significantly higher than those of whole blood as 452.7 $\pm$ 313.7 in PRP A2 ($p < 0.04$), and 687.9 $\pm$ 367 in PRP B2 ($p < 0.001$; Figure 1). The fold change after two kinds of double spin from 19 patients ranged from 1.1 to 6.3 times (Table 1).

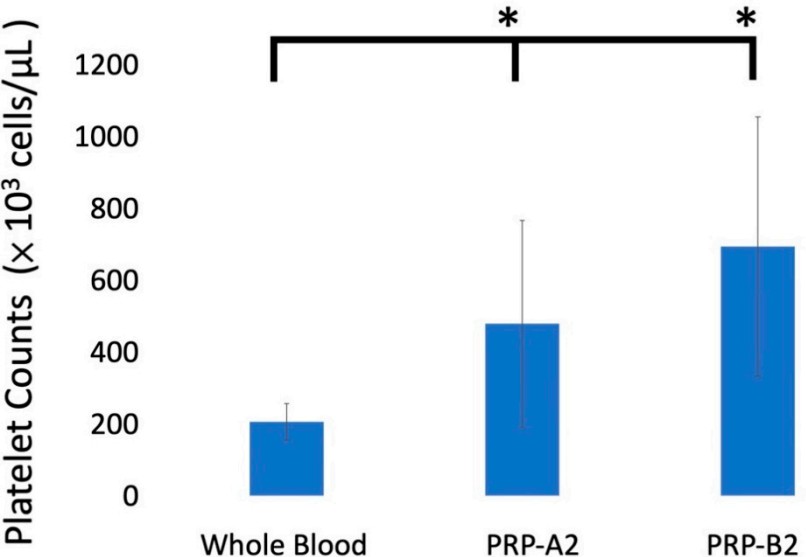

**Figure 1.** The platelet concentrations after two types of PRP preparations. The platelet concentrations were significantly higher than those of whole blood. * *p*-value < 0.05. Error bars showed mean ± standard deviation.

**Table 1.** Demographic data of the blood content from 19 patients.

|  |  | Whole Blood | PRP-A2 | PRP-B2 |
|---|---|---|---|---|
| Sex, M; F | 10; 9 |  |  |  |
| Age, year | 58.4 ± 16.3 |  |  |  |
| WBC, ($10^3$ cells/μL) |  | 5.4 ± 1.8 | 0.6 ± 0.5, (11%) | 1.6 ± 1.4, (29.4%) |
| RBC, (($10^6$ cells/μL)) |  | 4.6 ± 1.1 | 0.04 ± 0.03, (1%) | 0.1 ± 0.03, (2%) |
| HGB, (g/dL) |  | 13.8 ± 3 | 0.08 ± 0.04 (0%) | 0.1 ± 0.1, (0%) |
| HCT, ($10^3$ cells/μL) |  | 41.1 ± 8.4 | 0.22 ± 0.1, (1%) | 0.5 ± 0.2, (1.2%) |
| PLT, ($10^3$ cells/μL) |  | 205.3 ± 50.7 | 452.7 ± 313.7, (221%) | 687.9 ± 367, (335%) |

2.1.2. Time-Sequential Cytokine Release

**TGF-β1**

The TGF-β1 release at different time points (0 h, 24 h, 48 h, 72 h, and 96 h) in 3-chamber co-culture device was 4250.79 ± 1441.14, 4862.89 ± 1641.93, 4915.04 ± 1483.96, 5324.79 ± 1720.3, and 5390.04 ± 1371.79 pg/mL, respectively. They were 6052.67 ± 1035.28, 5999.44 ± 1040.79, 6225.11 ± 943.76, 6270.37 ± 1371.69, and 6005.04 ± 1011.19 pg/mL in 24-well plate (Figure 2a). There was a moderate to high correlation between TGF-β1 in co-culture device and 24-well plate until 96 h (Table 2).

**PDGF-AB**

The PDGF-AB release at different time points (0 h, 24 h, 48 h, 72 h, and 96 h) in 3-chamber co-culture device was 295.78 ± 44.53, 319.74 ± 89.34, 273.81 ± 82.14, 265.16 ± 77.69, and 267.85 ± 64.32 pg/mL, respectively. They were 296.99 ± 46.6, 341.84 ± 75, 308.48 ± 45.86, 334.76 ± 41.28, and 350.82 ± 47.21 pg/mL in 24-well plate (Figure 2b). There was a moderate to high correlation between PDGF-AB in a co-culture device and a 24-well plate for the first 24 h. Then it became modestly correlated from 48 to 96 h until 96 h (Table 2).

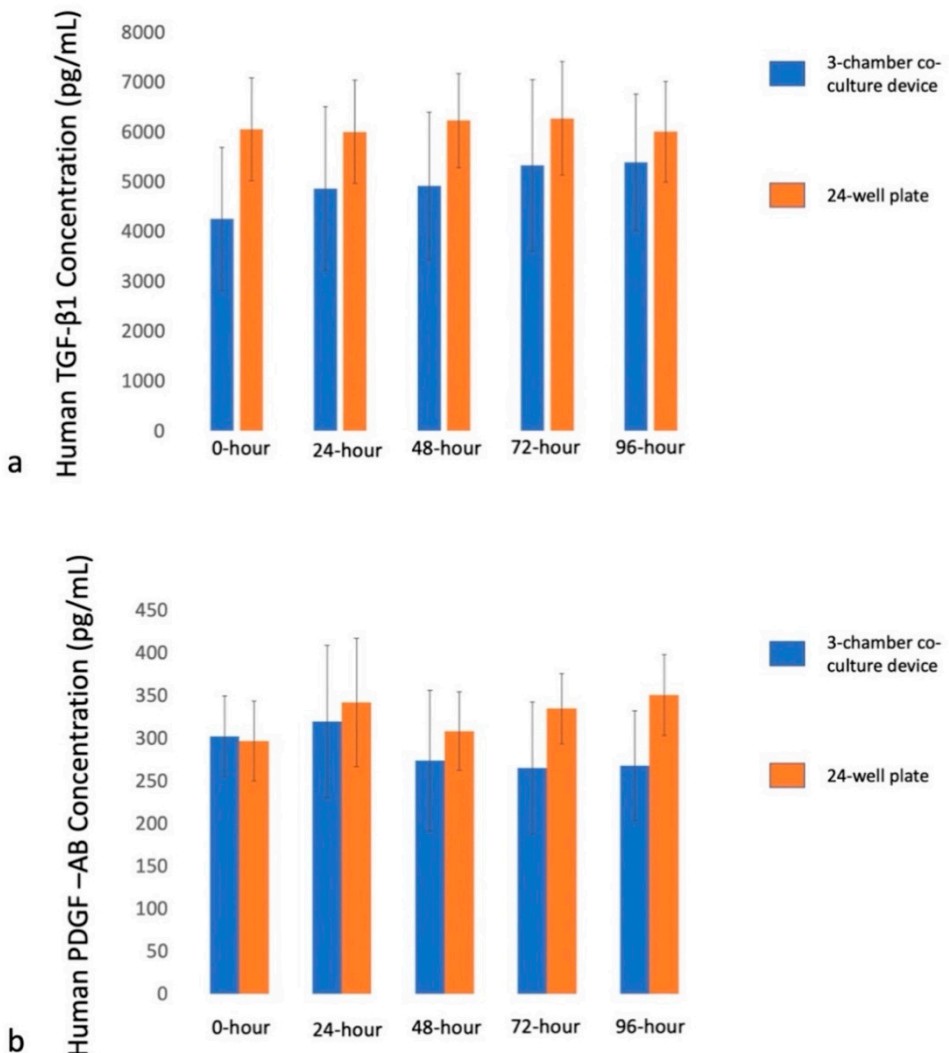

**Figure 2.** Cytokines release at different time points in 3-chamber co-culture device and conventional 24-well plate.
(**a**) TGF-β1 release at 0 h, 24 h, 48 h, 72 h and 96 h. (**b**) PDGF-AB release at 0 h, 24 h, 48 h, 72 h, and 96 h.

### 2.1.3. Correlation of Platelet Counts and Cytokine Concentration

Thirty-eight concentrations of platelets and cytokines (TGF-β1, PDGF-AB) with two different centrifugation methods were prepared.

**TGF-β1**

There was a weak-to-modest correlation between platelet and TGF-β1 concentration from 0 h to 72 h. Though not significant, they became high correlated at 96 h ([r] = 0.95, $p$ = 0.673; Table 3).

**PDGF-AB**

Negative and modest correlations were found between platelet and PDGF-AB concentrations thought all time points ([r] from −0.024 to 0.275), as Table 3 shows.

**Table 2.** The correlation between the concentration of TGF-β1 and PDGF-AB in different time points and culture devices.

| | | 0 h | 24 h | 48 h | 72 h | 96 h |
|---|---|---|---|---|---|---|
| TGF-β1 | Concentrations | | | | | |
| | 3-chamber co-culture | 4250.79 ± 1441.14 | 4862.89 ± 1641.93 | 4915.04 ± 1483.96 | 5324.79 ± 1720.3 | 5390.04 ± 1371.79 |
| | 24-well plates | 6052.67 ± 1035.28 | 5999.44 ± 1040.79 | 6225.11 ± 943.76 | 6270.37 ± 1371.69 | 6005.04 ± 1011.19 |
| | Pearson correlation coefficient between 2 devices | 0.682 | 0.759 | 0.737 | 0.825 | 0.763 |
| | *p*-value | <0.001 | <0.001 | <0.001 | <0.001 | <0.001 |
| PDGF-AB | Concentrations | | | | | |
| | 3-chamber co-culture | 295.78 ± 44.53 | 319.74 ± 89.34 | 273.81 ± 82.14 | 265.16 ± 77.69 | 267.85 ± 64.32 |
| | 24-well plates | 296.99 ± 46.6 | 341.84 ± 75 | 308.48 ± 45.86 | 334.76 ± 41.28 | 350.82 ± 47.21 |
| | Pearson correlation coefficient between 2 devices | 0.433 | 0.836 | 0.348 | 0.145 | 0.289 |
| | *p*-value | 0.094 | <0.001 | 0.187 | 0.592 | 0.277 |

**Table 3.** The correlation between the concentration of platelets and cytokines at different time points.

| | | 0 h | 24 h | 48 h | 72 h | 96 h |
|---|---|---|---|---|---|---|
| TGF-β1 | Pearson correlation coefficient | 0.139 | 0.21 | 0.1 | −0.14 | 0.95 |
| | *p*-value | 0.537 | 0.347 | 0.657 | 0.952 | 0.673 |
| PDGF-AB | Pearson correlation coefficient | −0.024 | 0.275 | 0.218 | 0.178 | 0.067 |
| | *p*-value | 0.93 | 0.303 | 0.418 | 0.51 | 0.804 |

## 3. Discussion

This study confirmed one of our hypotheses that there was a moderate to high correlation in TGF-β1 concentrations between the 3-chamber co-culture and conventional culture device until 96 h. However, the correlation of PDGF-AB concentrations in both devices had moderate to high correlation in the first 24 h, and then it became modestly correlated from 48 to 96 h. Regarding the relationship between platelet concentrations and the released cytokine concentrations, our hypothesis that the concentrations of the cytokine from PRPs at different time points would be correlated with platelet concentrations after production was rejected. A high correlation was only found between platelet and TGF-β1 concentration at 96 h. They were modestly correlated in other time points. Moreover, a negative or modest correlation was found between platelet and PDGF-AB concentration in all time points.

PRP was proved to stimulate the proliferation of several cell types in vitro, including osteoblasts [22,23], fibroblasts [24], tenocytes [25], chondrocytes [26], periodontal ligament cells [27], and bone mesenchymal stem cells [28]. It is not only used in the orthopedic field, but it is also commonly used to prevent and treat medication-related osteonecrosis of the jaws [29]. PRP provides a quick, minimally invasive, and relatively low-cost therapeutic strategy to enhance tissue healing because of the increased concentrations of autologous growth factors within [9]. However, there were mixed results in clinical use. For example,

Randelli and Gumina reported favorable outcomes and improved repair integrity when PRP was used in conjunction with rotator cuff repairs [7,8], while Castricini and Rodeo found no improvement in healing or functional score [30,31]. Sanchez et al. reported their results from the augmentation of platelet-rich fibrin matrices in surgically repaired Achilles tendon tears and concluded that PRP might enhance healing and functional recovery [32]. On the other hand, Schepull et al., in their Level 2, randomized controlled trial that included 30 patients in whom PRP was applied during Achilles tendon repair, found PRP was not useful for the treatment of Achilles tendon ruptures [33]. To maximize the advantages of PRP treatment, the optimal PRP formulation (leukocyte and platelet count, cytokines), platelet activation methods (thrombin, type 1 collagen, calcium gluconate), type of carrier vehicle (fibrin matrix, scaffold), and dosing regimen (single versus multiple injections, intraoperative versus delayed injection) [34]. Unfortunately, the best preparation protocols of PRPs, the precise cytokines combination, and the best timing to treat a specific disease at its different injury level are still unknown.

Cytokines within the $\alpha$-granules of platelets are believed to be responsible for tissue healing. Knowledge of the kinematics of cytokine release is fundamental to PRP application, especially the optimal timing and interval of injection of PRP in different kinds and stages of the disease. Oh et al. compared the cellular composition and cytokine-release kinetics of five different PRPs at 1 h, 24 h, 72 h, and 7 days after PRP preparation [13]. They found the fibroblast growth factor (FGF) and transforming growth factor (TGF) release quickly and decreased over time. On the other hand, platelet-derived growth factor (PDGF) and vascular endothelial growth factor (VEGF) were constantly released and sustained over 7 days. Therefore, the cytokine content was not necessarily proportional to the platelet count of the PRPs. In our study, we compared concentrations of TGF-$\beta$1 and PDGF-AB and at five time points after PRP preparation. The variations in the concentrations of different cytokines at different time points suggested that each type of cytokine release has individual dynamics. However, there are studies that reported a correlation between the platelet counts and the cytokine concentrations in PRP [35,36]. In our study, however, there was only a weak ($[r] = -0.084$) correlation of TGF-$\beta$1 and platelet concentration at 0 h. The correlation of PDGF-AB and platelet concentration was even negative ($[r] = -0.4$) at time zero. This corresponded to the results of Oh et al. [13] that the cytokine content was not necessarily proportional to the cellular composition of the PRPs.

The objective of using a 3-chamber co-culture device in PRP studies is because it works as a screening platform. The design could avoid the gelling effect of PRP [15,17] and provide more variables, e.g., diseased tissue or cells at different stages, when co-cultured with different PRPs. The relatively smaller culture area makes it possible to culture cells from scant specimens because sometimes the diseased tissue is very few. This helps clinicians to screen a tailored treatment to the extent of the disease process. For example, to find out the best PRP preparation for mild versus severely degenerative rotator cuff tear, only a few specimens are needed to be harvested during surgery. More types of PRPs could be co-cultured with cells from different tissues to see their potential to facilitate cell proliferation than in conventional culture wells with a larger diameter. This study confirmed the high correlation of cytokines in the 3-chamber co-culture device and conventional 24-well plate. This implies the 3-chamber co-culture device might replace conventional ones in further PRP studies.

The other findings of this study are the concentrations of TGF-$\beta$1 and PDGF-AB at specific time points. TGF-$\beta$1 is a key growth factor to stimulate collagen synthesis, cell proliferation, recruitment, and reduced scar formation in healing tendons [4]. PDGF-AB is an anabolic cytokine and had a role in inhibiting inflammation and pain as well as enhancing the biosynthesis of cartilage and bone matrix [37]. The best time point to start PRP treatment for a specific disease, however, is yet to be elucidated. In the current study, TGF-$\beta$1 and PDGF-AB showed consistent concentrations through 96 h after the preparation. These data may help clinicians to design individualized treatment protocols at different time points when PRP injection is considered.

There are still limitations regarding this study. First, the in vitro effects were not confirmed in all the in vivo studies because of many variables in the complex scenario where both PRPs and the lesion site might play a crucial role. This in vitro study implied that 3-chamber co-culture device might replace conventional culture well because the cytokines within both platforms had a high correlation with each other, at least for TGF-β1. Further study should focus on the cross-match of diseased tissue and the optimized PRP preparation and timing of usage accordingly. With the 3-chamber co-culture device, we could collect as much data regarding the different stages of a specific disease and its correlation with different kinds of PRPs. Second, the PRPs used in this study are not commercial ones. However, they shared similar preparation protocols as previously published results [38,39]. Third, only two cytokines were tested in this index study. There are many cytokines within platelets. More cytokines could have been tested if more blood was used in this index study.

## 4. Materials and Methods

This study was approved by the ethics committee by the authors' institute, and informed consent was obtained.

**PRP preparation**

Thirty milliliters of whole blood were drawn from each patient and then mixed with 1.25 mL anticoagulant solution (citrate phosphate dextrose adenine-1; CPDA-1) in a plain tube. The PRP was prepared by two double spin centrifugation processes using a bench-top centrifuge machine (Model: 5430R; Eppendorf, Germany). For the first double spin PRP preparation, 900 g for 5 min and then 1500 g for 15 min centrifugations were performed. The lower 3 mL of the plasma was termed PRP A2 (Figure 3) [39]. For the second double spin PRP preparation, centrifugation of 1500 rpm for 5 min and then 20 min at 6300 rpm were performed. Finally, half of the superficial plasma layer was removed, and the platelet pellet was suspended in the remaining half of the plasma volume termed PRP B2 (Figure 3) [38]. The platelet numbers of the PRPs were counted by an automatic hematology analyzer (Model: Sysmex XT-1800i; Kobe, Japan) after preparation.

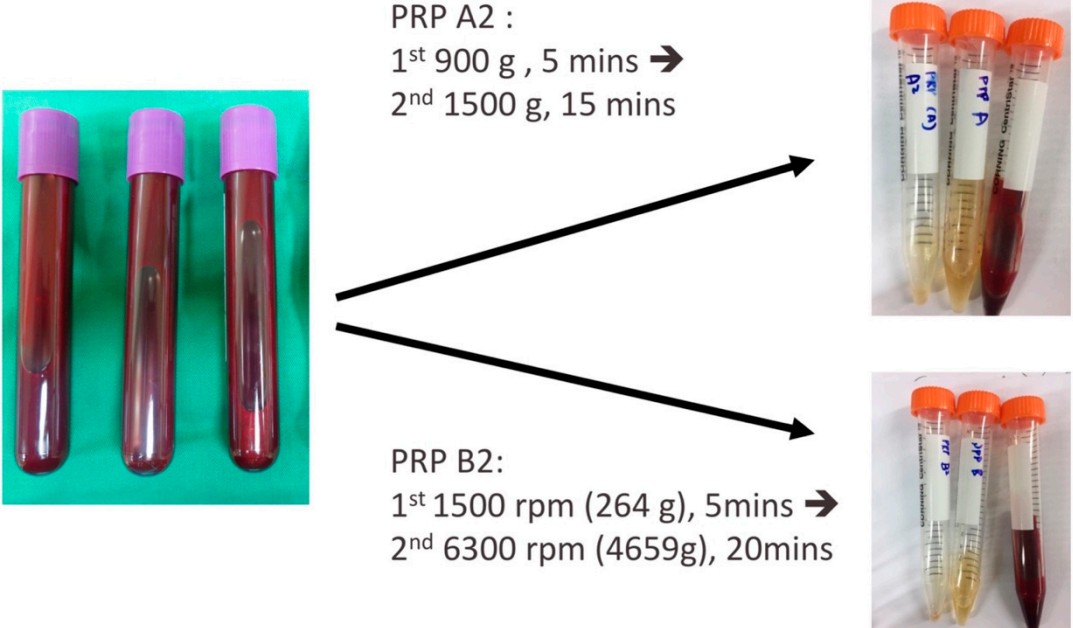

**Figure 3.** Two types of PRP preparations. PRP A2, 900 g for 5 min and then 1500 g for 15 min; PRP B2, 1500 rpm for 5 min and then 6300 rpm for 20 min.

**Evaluation of the cytokine composition in the 3-chamber co-culture device and the conventional 24-well culture plate at different time points**

To study the platelet-derived cytokines in 3-chamber co-culture device and conventional culture well (24-well plate; Figure 4), 40 μL PRP and 40 μL culture medium were added to one sub-chambers of the co-culture device (Figure 5a). Then, 100 μL of culture medium were added to each of the other two empty sub-chambers (Figure 5b). After gelling of the PRP overnight, 700 μL of culture medium were applied to the whole culture well to cause the culture medium to cross the central barrier (Figure 5c). The platelet-derived cytokines released in the medium would diffuse to all 3 sub-chambers, as shown in Figure 5d. The same procedure was performed in a 24-well plate as a control. After gelling of the PRP overnight, 900 μL of culture medium were applied to the 24-well culture well. The supernatant was collected at 0 h (immediately after culture medium exchange between each co-culture chamber), 24 h, 48 h, 72 h, and 96 h. TGF-β1 and PDGF-AB were analyzed by using a commercial immunoassay kit (Human TGF-β1 and Human PDGF-AB Quantikine® ELISA kit; R&D Systems, Minneapolis, MN, USA). The analytical protocol followed the manufacturer's instructions as previously described [40]. In addition, a serial dilution of the provided standard TGF-β1 and PDGF-AB solution was utilized to set up the calibration curve. Thus, the actual concentration of the TGF-β1 and PDGF-AB could be calculated based on the calibration curve. All samples were assayed in triplicate. The whole procedure was shown in Figure 6.

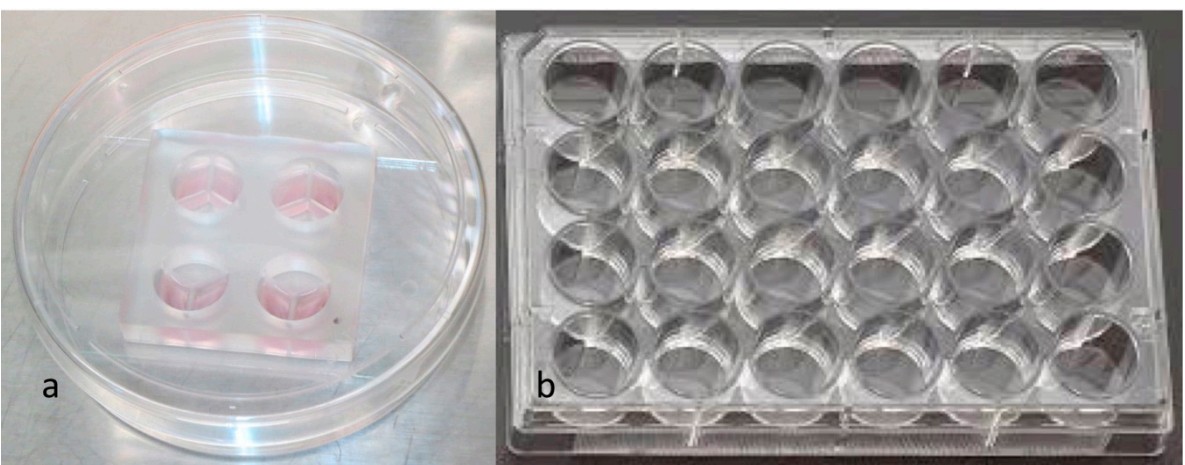

**Figure 4.** (**a**) 3-chamber co-culture device. (**b**) Conventional 24-well culture plate.

**Statistical Analysis**

Statistical analyses were performed with the SPSS software program (v 23.0; IBM Corp). The measured data are presented as the arithmetic mean and the standard deviation (SD). To compare platelet concentration between whole blood, PRP A2, and PRP B2, analysis of variance (ANOVA) followed by Bonferroni post hoc test was used. The concentrations of the cytokines in each incubation time were analyzed by Kruskall–Wallis test (nonparametric analysis of variance). A Bonferroni post hoc test was conducted to compare multiple values from each of the preparation conditions. Linear correlations between the platelet and cytokines content were analyzed in terms of the Pearson correlation coefficient. Correlations of cytokines between 3-chamber co-culture device and conventional 24-well plate at five time points were analyzed by the Pearson correlation coefficient. A *p*-value of <0.05 was considered significant.

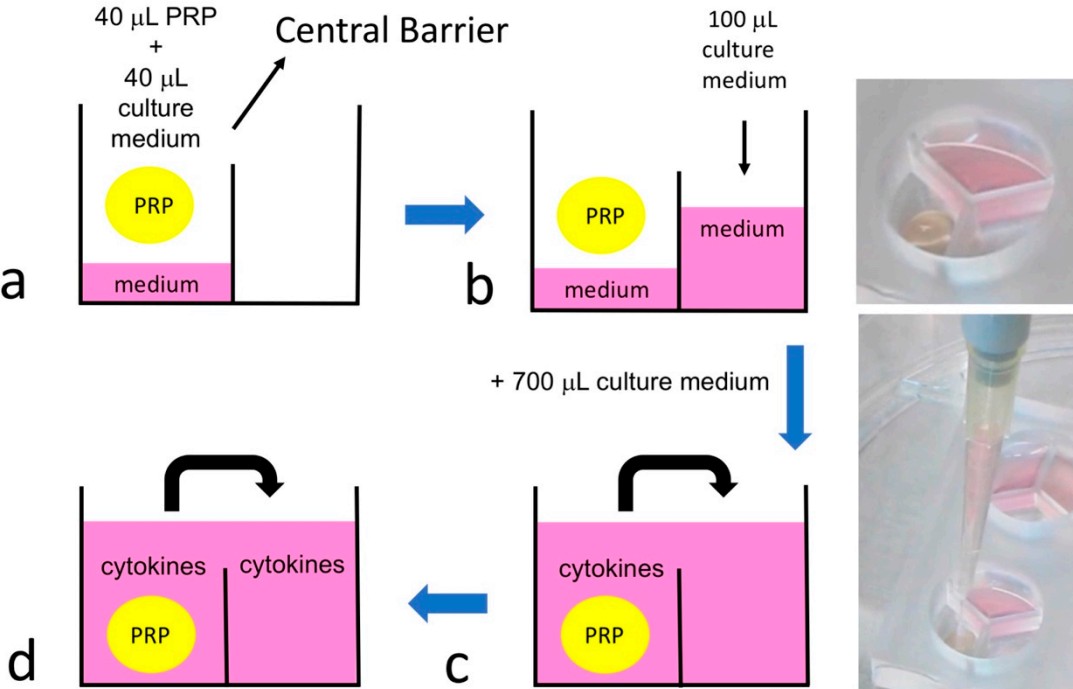

**Figure 5.** The idea of platelet-derived cytokines diffusion in the 3-chamber co-culture system. (**a**) 40 μL PRP and 40 μL culture medium were added to one sub-chambers of the co-culture device. (**b**) Then, 100 μL culture medium were added to each of the other two sub-chambers. (**c**) After gelling of the PRP overnight, 700 μL culture medium was added to the whole culture well to cause the culture medium to cross the central barrier. (**d**) The platelet-derived cytokines released in the medium would diffuse to all 3 sub-chambers.

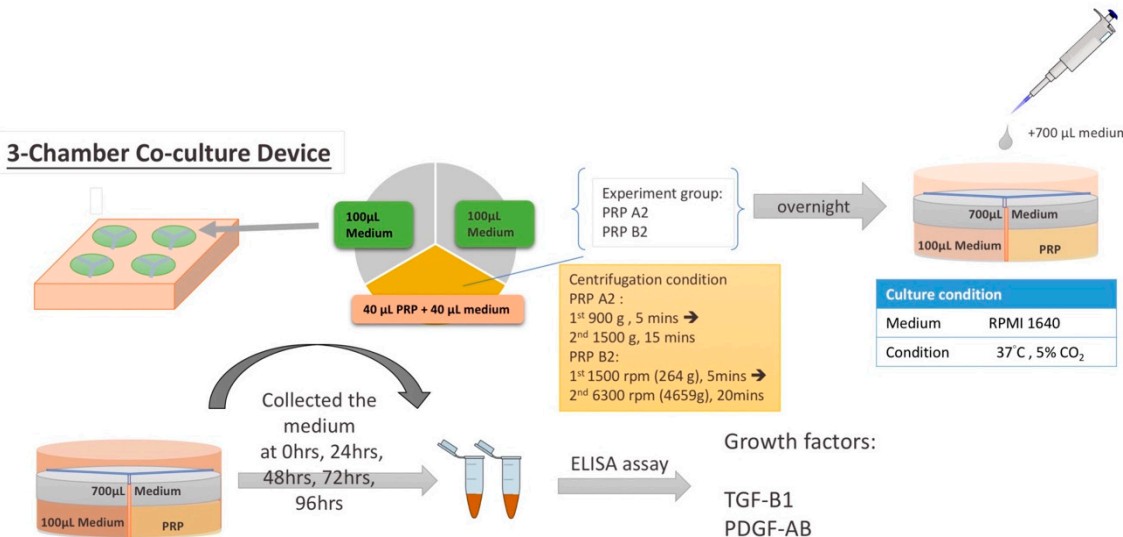

**Figure 6.** The whole procedure of this study.

## 5. Conclusions

TGF-β1 and PDGF-AB revealed a time-dependent manner of release at five time points after specific PRP preparation protocols. There is a moderate to high correlation of the TGF-β1 and PDGF-AB concentration in both devices at different time points. However, TGF-β1 and PDGF-AB concentrations are not always proportional to the platelet concentration of

the PRPs. These results may contribute to the optimization of PRP preparation and the exact timing of usage in clinical application.

**Author Contributions:** C.-H.C. created the ideas, formulation, and wrote the paper. P.C. performed the statistics. K.F.L. provided the technical support of co-culture design. A.C.-Y.C. provided the PRP. Y.-S.C. and K.-Y.H. shared their experience in PRP preparation. All authors have read and agreed to the published version of the manuscript.

**Funding:** This research was funded by Taiwan Minister of Science and Technology, grant number "MOST 107-2314-B-182A-150 -MY3", and Chang Gung Memorial Hospital, grant number "CM-RPG5K0091". The APC was funded by "Chang Gung Memorial Hospital".

**Acknowledgments:** The authors gratefully thank Taiwan Minister of Science and Technology and Linkou Chang Gung Memorial Hospital for financial support of this study (Grant: MOST 107-2314-B-182A-150 -MY3, CMRPG5K0091).

**Conflicts of Interest:** The authors declare no conflict of interest.

## Abbreviations

| | |
|---|---|
| PRP | Platelet-rich plasma |
| TGF-β1 | transforming growth factor beta-1 |
| PDGF | platelet-derived growth factor |
| SD | standard deviation |
| FGF | fibroblast growth factor |
| TGF | transforming growth factor |
| PDGF | platelet-derived growth factor |
| VEGF | vascular endothelial growth factor |

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
