# Peer review of "Time-Dependent Cytokine-Release of Platelet-Rich Plasma in 3-Chamber Co-Culture Device and Conventional Culture Well"

_applsci, doi:10.3390/app11156947_

Round 1
Reviewer 1 Report
In the original article by Dr Chih-Hao Chiu et al. entitled „Time-Dependent Cytokine-Release of Platelet-Rich Plasma in 3-chamber Co-culture Device and Conventional Culture Well”, the Authors investigated the effect of novel methodological approach to obtain platelet-rich plasma (PRP) on the cytokine level in the PRP. In my opinion, the paper reports interesting results however the quality of the presentation is very low and needs to be improved. My specific comments are listed below.
Table 1. It is unclear what is the meaning of the numbers in parentheses in last two columns. It should be stated in the legend below the table. I suspect that these numbers are percentages of cell count parameters calculated in respect to whole blood. If so, they are sometimes calculated erroneously – e.g. BASO (whole blood) 0.04; BASO PRP-A2 0.03 – should be 75% not 7.4%. BASO PRP-B2 0.01 should be 25%, not 42%.
Table 1. The units in the headline of the table are wrong. RBC in the blood count is always given in 106 not 103. HCT is given as a percentage, HGB is given in g/dl. The table 1 should be corrected and I advise the authors to check all the calculations.
Figure 1. What is represented by the error bars on the figures? Standard deviation? Standard error of the mean? It should be clarified in the figure legend.
Table 2. Given p-value=0 means that the Authors do not understand what the p-value is. It by definition cannot be 0. Please correct this.
Correlations shown in the text (lines 128-129; 132-133) and in the Table 2 and 3. The Authors claim that the correlations were “low to modest” or “modest” or “high” whereas the p-values were insignificant. I would like to mention that if no significance is reached, the strength of correlation does not matter since it could be achieved by a pure chance.
Whole text of the manuscript: Please use dots after the references in square brackets, not before.
e.g. line 44: tissue healing are released.[1-3] -> tissue healing are released [1-3].
Line 57 methods[12]. -> methods [12].
Line 81 “19 patients (10 women, 11 men)” the numbers in parenthesis do not sum up, please correct.
Line 90 fin –> in
Line 90 p=0.032 –> p<0.04 (it is common practise to round the numer of p to one significant digit)
Line 90 p=0 (its is impossible for p=0. It would mean that the result is “sure”. Mayby the was very low, e.g. p<10-5; if so it should be presented here)
Author Response
Table 1. It is unclear what is the meaning of the numbers in parentheses in last two columns. It should be stated in the legend below the table. I suspect that these numbers are percentages of cell count parameters calculated in respect to whole blood. If so, they are sometimes calculated erroneously – e.g. BASO (whole blood) 0.04; BASO PRP-A2 0.03 – should be 75% not 7.4%. BASO PRP-B2 0.01 should be 25%, not 42%.
Reply: Thanks for the comment. Yes, they are indeed the percentages of cell count parameters calculated in respect to whole blood. The errors are corrected in the revised manuscript.
Table 1. The units in the headline of the table are wrong. RBC in the blood count is always given in 106 not 103. HCT is given as a percentage, HGB is given in g/dl. The table 1 should be corrected and I advise the authors to check all the calculations.
Reply: The units in Table 1 were revised and the calculations were performed again. Thanks for the comment.
Figure 1. What is represented by the error bars on the figures? Standard deviation? Standard error of the mean? It should be clarified in the figure legend.
Reply: The error bars showed mean ± standard deviation. It was corrected in the revised manuscript. Thanks for the comment.
Table 2. Given p-value=0 means that the Authors do not understand what the p-value is. It by definition cannot be 0. Please correct this.
Reply: The “p-value=0” in Table 2 were corrected into “p-value < 0.001” in the revised manuscript. Thanks for the comment.
Correlations shown in the text (lines 128-129; 132-133) and in the Table 2 and 3. The Authors claim that the correlations were “low to modest” or “modest” or “high” whereas the p-values were insignificant. I would like to mention that if no significance is reached, the strength of correlation does not matter since it could be achieved by a pure chance.
Reply: Yes, that’s why we mentioned in the manuscript that “Though not significant” in Line 129. Thanks for the comment.
Whole text of the manuscript: Please use dots after the references in square brackets, not before.
e.g. line 44: tissue healing are released.[1-3] -> tissue healing are released [1-3].
Line 57 methods[12]. -> methods [12].
Line 81 “19 patients (10 women, 11 men)” the numbers in parenthesis do not sum up, please correct.
Line 90 fin –> in
Line 90 p=0.032 –> p<0.04 (it is common practise to round the numer of p to one significant digit)
Line 90 p=0 (its is impossible for p=0. It would mean that the result is “sure”. Mayby the was very low, e.g. p<10-5; if so it should be presented here)
Reply: They are all corrected in the revised manuscript. Thanks for the comment.

Reviewer 2 Report
The manuscript submitted to Applied Sciences entitled “Time-Dependent Cytokine-Release of Platelet-Rich Plasma in 3-chamber Co-culture Device and Conventional Culture Wells” is an original article which aim to evaluate cytokine release of Platelet-Rich Plasma (PRP) in different culture systems.
On my opinion the article is interesting, but some improvements are needed.
- English language: minor spell check required.
- Abstract: Please structure the abstract to attract the reader's attention,and to adapt it accordingly
- Introduction: I would suggest the improvement of this section.
For example, on page 2 line 48 please include suggested text and reference:
<<PRP is used to promote wound healing, although its use still raises various controversies related to the effectiveness of platelet concentrates in tissue regeneration [https://doi.org/10.1007/s00784-020-03702-w].>>
- Results: This section has been properly prepared.
- Discussion: Please discuss use of PRP in prevention and treatment of medication-related osteonecrosis of the jaws with reference to a systematic review published in the EACMFS reference journal [https://doi.org/10.1016/j.jcms.2020.01.014].
- Materials and Methods: This section needs improvements. Please insert a subsection in the beginning regarding protocol. Was the study approved by an ethics committee?
- Conclusion: This section has been properly prepared.
- Figures: Please improve quality and resolution.
- Summary of abbreviations required at the end of the manuscript prior to “Reference” section.
These changes are required to make the manuscript suitable for publication.
Thanks for the opportunity to review this manuscript.
Author Response
On my opinion the article is interesting, but some improvements are needed.
Introduction: I would suggest the improvement of this section.
For example, on page 2 line 48 please include suggested text and reference:
<<PRP is used to promote wound healing, although its use still raises various controversies related to the effectiveness of platelet concentrates in tissue regeneration [https://doi.org/10.1007/s00784-020-03702-w].>>
Reply: The description was included in the revised manuscript (Ref. 6). Thanks for the comment.
Discussion: Please discuss use of PRP in prevention and treatment of medication-related osteonecrosis of the jaws with reference to a systematic review published in the EACMFS reference journal [https://doi.org/10.1016/j.jcms.2020.01.014].
Reply: The description was included in the revised manuscript (Ref. 29). Thanks for the comment.
Materials and Methods: This section needs improvements. Please insert a subsection in the beginning regarding protocol. Was the study approved by an ethics committee?
Reply: This study was approved by the ethics committee by the authors’ institute, and informed consent was obtained. Thanks for the comment.
Summary of abbreviations required at the end of the manuscript prior to “Reference” section.
Reply: A summary of abbreviations was provided in the revised manuscript. Thanks for the comment.

Round 2
Reviewer 1 Report
The manuscript has been improved and the majority of errors have been corrected. I have no further comments.
This manuscript is a resubmission of an earlier submission. The following is a list of the peer review reports and author responses from that submission.
Round 1
Reviewer 1 Report
Chiu et al. reported the measurement of platelets and cytokines in platelet-rich plasma in 3-chamber co-culture devices. This study showed a potential to use 3-chamber co-culture device rather than conventional 24-well culture plate to save the material in evaluating the effect of platelet-rich plasma. However, as reported by the authors in Page 6 of the manuscript, there are a number of limitations in the study. The biggest drawback is the limited number of sample size, where only 4 patients were evaluated. This led to weak correlation coefficient and more importantly the lack of statistical significance in most time-points evaluated (Table 2). Thus, it is difficult to draw any conclusions regarding the correlation of platetlet concentrations and cytokine levels or the correlation of 3-chamber co-culture device and 23-well setup. The authors need to expand the sample size and repeat the measurements to achieve statistically significant results. Further, there are a number of instances where the sentences were broken. The authors need to address the scientific issues as well as language editing.
Reviewer 2 Report
In the present study, Chiu and colleagues compare normal culture plates with a 3-chamber device for co-culture approaches with platelet-rich plasma in terms of tissue regeneration after surgery or injury. In this manner, the authors compare concentrations of TGF beta 1 and platelet-derived growth factor AB by ELISA at five different time points in both devices and found that concentrations of both cytokines were comparable.
General concerns: The advantage of the evaluated 3 chamber device is absolutely not clear in comparison to a normal plate. The whole manuscript contains data on platelet counts of 4 different subjects and ELISA data on TGF beta 1 and platelet-derived growth factor AB released from platelets. The reaming figures contain drawings (Fig. 5 and 6) or photos of cell culture plates or falcons filled with blood or PRP. That kind of figure does not add any further information. Additionally, the manuscript contains a lot of typos. Furthermore, in the discussion section, the authors propose a co-culture of cells for their 3-chamber device with PRP. Why didn´t the authors perform such a co-culture approach with cells as proof of concept to show that their 3-chamber device is superior to normal cell culture plates or at least comparable? Thus, this reviewer has the impression that the authors exaggerate their whole story by showing less and meaningless data embellish with some photos and drawings. Thus, this manuscript should be rejected.
Here are some minor points:
Introduction: Please spend some sentences on the indications or diseases in which PRP is applied to patients. Additionally, the exact application and usage of the 3-chamber device do not become clear in the introduction. There must be provided more detailed information on the aim of the study to make comprehensible to the readership. Line 21: culture plates Line 21: time-points Line 24: cytokine and not cytokines Line 56: the main idea of the system is Line 66: cytokine and not cytokines Line 66: wound? Do you mean would? Line 69: time-points and not time point You are talking about 3 chamber system but you show a 2 chamber drawing in figure 5 to explain the mechanism of your system. Where is the need for the third chamber? Results: the description of the subjects should appear in the Materials and Methods section and not in the results. Line 81: The platelet concentrations after the second spin were 709 ± 353.1. This is not a concentration. Do you mean per µL or mL? Please indicate. Line 124: full stop is missing. There is no reference to table 1 in the results. Please add. Table 1: whoe blood. Do you mean whole blood? Correct! Table 1: why do you give so many decimal places for the fold column? That makes no sense. Figure 1: where are the differences in the 2 types of PRP preparation? Please explain in the text in the corresponding section and not later in figure 3.